# Effects of 10 Dwarfing Interstocks on Cold Resistance of 'Tianhong 2' Apple

**Junli Jing, Mingxiao Liu, Baoying Yin, Bowen Liang, Zhongyong Li, Xueying Zhang, Jizhong Xu *** **and Shasha Zhou ***

College of Horticulture, Hebei Agricultural University, Baoding 071001, China; 18730277283@163.com (J.J.);
liumingxiao0518@163.com (M.L.); by40901123@126.com (B.Y.); lbw@hebau.edu.cn (B.L.);
yylzy@hebau.edu.cn (Z.L.); zhangxueying1996@163.com (X.Z.)
* Correspondence: xjzhxw@126.com (J.X.); zhoushasha198609@126.com (S.Z.)

**Abstract:** The lack of dwarf stock with good cold resistance has affected the production of apples in northern China. Annual dormant branches of 'Tianhong 2' apple were grafted on 10 different dwarf interstocks (the rootstocks were the seedlings of *Malus hupehensis* var. *Pingyiensis*) as test materials. Among these 10 interstocks, Huang 6, 244, NO.1, 53, 24-5, ZC9-3, Jizhen 1 were newly developed by us (Apple Research Group of Hebei Agricultural University), and three interstocks with different degrees of cold resistance (GM256–with strongest cold resistance, SH40–with stronger cold resistance, M9–with poor cold resistance) were used as controls. The semi-lethal temperature (LT50) and related physiological indexes of the branches in the overwintering process were studied. Based on the comprehensive physiological indexes, the effects of 10 interstocks on cold resistance of the 'Tianhong 2' apple were analyzed. The results showed that the effects of 10 kinds of interstocks on the cold resistance of 'Tianhong 2' apple were quite different. The order of effects on cold resistance from strong to weak was as follow: GM256 > Huang 6 > 244 > NO.1 > 53 > 24-5 > ZC9-3 > Jizhen 1 > SH40 > M9. The purpose of this study was to screen out the interstocks with strong cold resistance, in order to provide some basis for the selection and utilization of interstocks with strong cold resistance in apple cultivation to further promote the development of the apple industry in China.

**Keywords:** apple; dwarfing interstocks; cold resistance

## 1. Introduction

The apple (*Malus domestica* Borkh) is a nutritious fruit and is consumed worldwide, providing an important food source [1]. China is in a dominant position in apple production globally with both the largest apple growing area and the largest export of fresh apples [2]. However, in many apple-producing areas of China (such as high-latitude and high-altitude areas), there are different degrees of low-temperature freezing injury. Freezing injury affects the growth, development, and yield of apple trees, and can even lead to tree death, thus posing a major threat to the development of the apple industry [3]. Low-temperature stress is a key limiting factor affecting plant growth, physiology, and metabolism, affecting overall yields [4,5]. Therefore, it is particularly important to improve the cold resistance of apple trees. Rootstock is an important factor in conferring apple cold resistance [6]. Grafting desirable apple varieties on rootstocks with inherent strong cold resistance can improve the cold resistance of the mature fruit trees. In the northern apple-producing areas of China, the improper application of dwarf rootstock has led to poor cold-resistant varieties with frequent occurrence of freezing injury [7]. Therefore, the cultivation of apple cold-resistant dwarf rootstocks has attracted much attention. Screening dwarf rootstocks with strong cold resistance is necessary to develop superior varieties to alleviate and prevent overwintering freezing injury of apple trees. Evaluating and identifying the mechanism of cold resistance in apple rootstocks and breeding dwarf rootstocks with strong cold resistance, suitable for northern China, are necessary to promote the development of the apple industry and

expand apple cultivation. To do this, we have bred multiple rootstocks with dwarfing potential, and evaluated their various characteristics, including cold resistance.

'Tianhong 2' (*Malus domestica* Borkh. cv. 'Tianhong 2') apple is a short-branch Fuji variant (*Malus domestica* Borkh. cv. Fuji). This variety was bred by the Apple Research Group of Hebei Agricultural University in 1994 and certified as a new variety by the Forest Tree Variety Approval Committee of Hebei Province, China in 2005 [8]. The fruit of 'Tianhong 2' is round and the fruit shape index can reach more than 0.9. The average single fruit weight is approximately 260 g. The fruit surface is smooth, and the surface color is bright red. Its pulp is yellow-white, crispy, and juicy. The taste is sweet and sour. The fruit contains 14.5%~16.8% soluble solids and has good aroma. On the whole, the quality is superior. The fruit matures between late October and early November in Baoding, Hebei Province, China and can be stored for 13 months. This variety is widely cultivated in Hebei Province, especially in Baoding (located in northern China). In this area, fruit growers usually choose vigorous rootstocks adapted to local climate as the base stock to improve the comprehensive resistance of 'Tianhong 2' apple trees, and graft dwarf interstocks on the base stock to dwarf the trees. Selecting suitable, highly cold-resistant dwarfing interstock will improve the yield and quality of the 'Tianhong 2' apple.

In this study, annual dormant branches of 'Tianhong 2' apple were grafted onto 10 different interstocks and tested for cold resistance. The tested interstocks include seven newly developed rootstocks with dwarf potential that were cultivated by the Apple Research Group of Hebei Agricultural University: Jizhen 1, Huang 6, 244, NO. 1, 53, 24-5, and ZC9-3. Three other interstocks that are widely cultivated in China were used as controls: SH40 and GM256, which exhibit strong cold resistance, and M9 with poor cold resistance. All interstocks were grafted on apomictic *Malus hupehensis* var. *Pingyiensis* as the base rootstock.

Semi-lethal temperature (LT50) is an important index widely used to evaluate plant cold resistance [9–11]. The cold resistance of apples may be closely related to many indicators. For example, reactive oxygen species such as superoxide anion are harmful substances and are produced at high levels in plants under low-temperature stress [12]. SOD and POD are important protective enzymes that can remove reactive oxygen species as part of plant cold resistance [13]. Proline, soluble sugar, and soluble protein are osmotic regulators that are essential players in the process of plant cold resistance [14]. Starch [15] and malondialdehyde (MDA) [16] are also important indicators of cold resistance of plants. Because many factors affect the cold resistance of fruit trees, it may be insufficient to measure cold resistance using a single index. Principal component analysis can be used to comprehensively evaluate cold resistance using multiple indicators. This type of analysis has been successfully applied to assess cold resistance of apples [7], grapes [9], and other crops. In recent years, there have been many reports on the cold resistance of apples. Some studies utilized apple rootstocks grown in an artificial low-temperature environment to measure different indicators and evaluate cold resistance [7,17]. Other studies measured cold resistance-related indicators to assess cold resistance in apple rootstocks subjected to natural overwintering and low temperature [7].

In this study, the semi-lethal temperature (LT50) and physiological indexes related to cold resistance were measured by conductance, and the different responses of various indexes to low-temperature stress were studied, which provided theoretical reference for studying the physiological characteristics of plant cold resistance during the process of overwintering. Principal component and cluster analysis were used to comprehensively evaluate the effects of cold resistance of these 10 different interstocks to 'Tianhong 2' apple. Selecting some new interstocks with strong cold resistance can provide some certainty in the choice and use of interstocks with strong cold hardiness for apple cultivation, further promoting the development of the apple industry. In many apple production areas, freezing injury caused great economic losses for the industry, and the hardiness of rootstock and interstock has a great influence on the cold resistance of apple tree. Therefore, this research has great significance for the development of apple industry.

## 2. Materials and Methods

### 2.1. Plant Materials

The test materials were grown in the West Campus of Hebei Agricultural University, Baoding City, Hebei Province, China. At the end of March 2019, the annual seedlings of *Malus hupehensis* var. *Pingyiensis* (apomixis) with the same growth vigor were planted in pots (22 cm high, upper diameter 30 cm, and lower diameter 20 cm) as the base rootstocks, and the no budding buds of 10 dwarfing interstocks were grafted on the base rootstock in early April 2019 by the plate budding method. The 10 dwarfing interstocks included: SH40 (with strong cold resistance), GM256 (with strongest cold resistance), and M9 (with poor cold resistance), which are widely used in production and were used as controls in this experiment; Jizhen 1, 53, 244, 24-5, Huang 6, ZC9-3, and NO.1, which were new dwarfing rootstocks cultivated by the Apple Research Group of Hebei Agricultural University. The background information of these seven new rootstocks was shown in Table 1. The no budding buds of 'Tianhong 2' apple were grafted on the interstock in early April 2020. Under unified management, 30 potted trees with the same growth vigor were selected from each rootstock-interstock-scion combination for the experiment. The annual dormant branches of 'Tianhong 2' (the scion) grafted on the 10 interstocks were used as test materials in the winter of 2020–2021.

**Table 1.** Background information of the seven new rootstocks.

| Rootstock | Parents | Breeding Institution |
|---|---|---|
| Jizhen 1 | seedling progeny of SH40 | |
| 53 | seedling progeny of SH40 | |
| 244 | seedling progeny of SH40 | |
| NO. 1 | seedling progeny of SH40 | Hebei Agricultural University |
| 24-5 | *Malus micromalus* × Inner Mongolia11 | |
| Huang 6 | seedling progeny of *Malus robusta* Rehd | |
| ZC9-3 | seedling progeny of P22 | |

### 2.2. Test Methods

#### 2.2.1. Materials, Treatment and Sampling

The branch samples were taken on 22 October 2020, 1 December 2020, 5 January 2021, and 5 March 2021. The maximum temperature, minimum temperature, and average temperature were recorded on each sampling day (shown in Table 2). Annual dormant branches (about 0.5 cm in diameter) of 'Tianhong 2' that were disease-free, strong, and exhibited consistent growth were selected for sampling. The harvested branches were divided into two parts for every rootstock-interstock-scion combination. One part was cleaned immediately after harvest and the branch bark was removed, quickly frozen in liquid nitrogen, and stored in a −80 °C freezer before the determination of physiological and biochemical indexes. The other part was cleaned with distilled water and deionized water and then divided into six groups and transferred to six plastic sealing bags (sprayed with deionized water to prevent the branches from drying out) for separate artificial low-temperature treatments (Table 3).

**Table 2.** Temperature on each sampling day during overwintering.

| Sampling Date | Natural Wintering Temperature/ °C | | |
|---|---|---|---|
| | Maximum Temperature | Minimum Temperature | Average Temperature |
| 22 October 2020 | 18 | 2 | 9.46 |
| 1 December 2020 | 1 | −5 | −1.08 |
| 5 January 2021 | −2 | −14 | −7.33 |
| 5 March 2021 | 10 | 1 | 4.29 |

**Table 3.** Cold treatments for cold resistance determination.

| Sampling Date | Temperature Settings/ °C | | | | | |
|---|---|---|---|---|---|---|
| 22 October 2020 | 4 | −8 | −20 | −26 | −36 | −45 |
| 1 December 2020 | 4 | −8 | −22 | −36 | −50 | −60 |
| 5 January 2021 | 4 | −10 | −22 | −45 | −55 | −65 |
| 5 March 2021 | 4 | −12 | −20 | −28 | −36 | −45 |

The low-temperature treatment was performed as described by Wilner [18]. A variable temperature, ultra-low-temperature refrigerator was used for the artificial freezing treatments, with six temperature gradients included the temperature at which all samples are alive and the temperature at which all samples are killed (Table 3). The cooling rate of the refrigerator was 6 °C/h, and samples were placed for 12 h after the temperature reached the set temperature. After that, the samples were thawed at 0 °C for 8 h, and then thawed at 4 °C for 24 h. After low-temperature treatment, the relative conductivity of the branches was measured.

Each treatment was performed with three repetitions. After low-temperature treatment, the relative conductivity was determined and the LT50 value was calculated.

2.2.2. Relative Electrolyte Leakage (REL) Measurement and LT50 Calculation

The relative electrolyte leakage (REL) and LT50 were measured and calculated according to the methods of Wilner [18] and Zhang [19], with slight modifications. The low-temperature treated branches were cut into 15 mm branch sections, split in the cross-sectional direction, and put into a test tube containing 10 mL ultrapure water. The test tubes were shaken on a shaking table for 24 h. After shaking, a DDS-307A conductivity meter (Shanghai Yidian Scientific Instrument Limited Liability Company, Shanghai, China) was used to measure the initial conductivity R1 and the blank conductivity R0. Tubes were then incubated in a boiling water bath for 30 min, shaken for 24 h, and then the final conductivity R2 was measured. The relative conductivity E of each treatment was calculated according to the formula E (%) = (R1 − R0)/(R2 − R0) × 100. The treatment temperatures and relative conductivity were fitted with a logistic equation using SPSS 21.0 software to calculate the inflection point temperature, which is the LT50 (C in the following Formula (1)).

$$y = \frac{A}{1 + e^{B(C-X)}} + D \tag{1}$$

In this formula (1). y = relative electrolyte leakage rate (%); X = low-temperature treatment temperature (°C); A is the difference between the highest E and the lowest E, B = slope (%•°C), the slope of the curve at the inflection point temperature; C = inflection point temperature (°C); and D is the lowest value of E.

2.2.3. Determination of Relevant Physiological Indexes

The content of MDA was determined by the thiobarbituric acid (TBA) method [20]; the content of superoxide anion was determined by the hydroxylamine oxidation method [21]; SOD activity was determined by the nitroblue tetrazolium (NBT) photoreduction method [20]; POD activity was determined by the guaiacol colorimetry method [20]; the content of soluble sugar and starch were determined by anthrone colorimetry [22]; the content of soluble protein was determined by Coomassie brilliant blue G-250 staining [23]; and the content of free proline was determined by acid ninhydrin staining [21]. Refer to the corresponding literature for specific steps.

2.2.4. Statistical Analysis

The data were fitted by SPSS 21.0 software using a logistic equation to calculate the LT50 values. Microsoft Excel 2010 software was used to analyze the test data and draw the figures. SPSS 21.0 was used for one-way ANOVA, and the Duncan test ($p < 0.05$) was used

for significance analysis. SPSS 21.0 was used for descriptive analysis, principal component analysis, and cluster analysis.

## 3. Results

*3.1. Changes of LT50 of 'Tianhong 2' Apple Branches Grafted on Different Interstocks under Natural Overwintering Process*

After the different low-temperature treatments, the relative conductivity of the apple branches grafted on interstocks was measured. The values were fit to a logistic equation to determine the LT50 values for each interstock, as shown in Figure 1. During overwintering stress, the LT50 changes corresponded to the change trend of winter temperature, which decreased firstly and then increased. From 22 October 2020 to 5 January 2021, with the gradual decrease in winter temperature, after adapting to the changes of external cold stress environmental conditions, the cold resistance of the branches grafted on each interstock increased and the corresponding LT50 values decreased, so the lowest LT50 values were obtained in January when the outside air temperature was the lowest and the stress was the most serious. From 5 January 2021 to 5 March 2021, as the temperature increased, the LT50 vales also increased. During overwintering, the branches of 'Tianhong 2' apple grafted on GM256 and Huang 6 showed the strongest cold resistance.

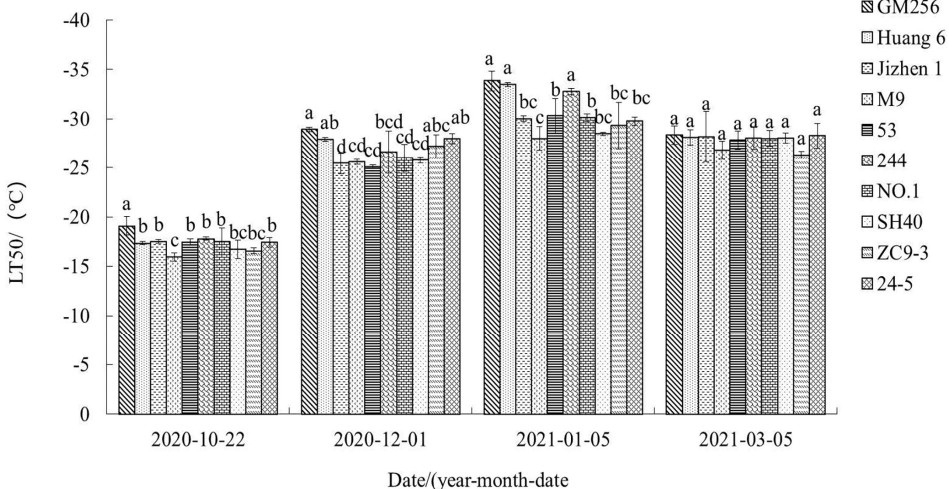

**Figure 1.** LT50 values of 'Tianhong 2' apple branches grafted on different interstocks. Note: In the figure, the lowercase letters indicate that significant differences exist between branches grafted on different interstocks ($p < 0.05$).

*3.2. Changes of Physiological Indexes of 'Tianhong 2' Apple Branches Grafted on Different Interstocks under Natural Overwintering Process*

3.2.1. Change of MDA Content

As shown in Figure 2, the change of MDA content in the branches was opposite to the change trend of winter temperature, first increasing and then decreasing. From 22 October 2020 to 5 January 2021, with the decrease in winter temperature and the intensification of low-temperature stress, the MDA content of apple branches increased, and MDA content was highest when the temperature stress was the most serious in January. From 5 January 2021 to 5 March 2021, as temperatures increased, low-temperature stress was alleviated, and the MDA content was reduced. Throughout overwintering, MDA levels remained lower in 'Tianhong 2' branches grafted on GM256 and Huang 6 and relatively higher for branches of 'Tianhong 2' grafted on ZC9-3.

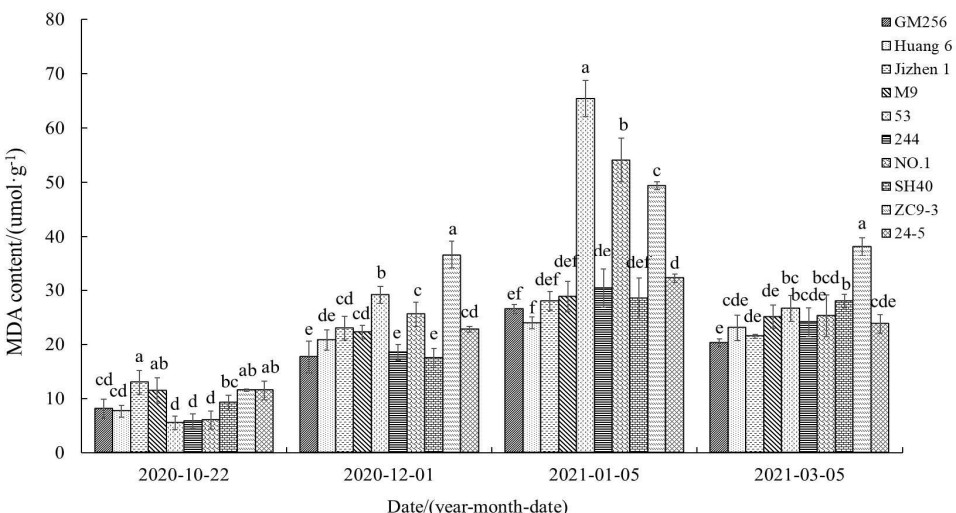

**Figure 2.** MDA content in 'Tianhong 2' apple branches grafted on different interstocks. Note: In the figure, the lowercase letters indicate that significant differences exist between branches grafted on different interstocks ($p < 0.05$).

### 3.2.2. Change of Superoxide Dismutase (SOD) Activity

As shown in Figure 3, the change of SOD activity in 'Tianhong 2' apple branches grafted on different interstocks was opposite to the change trend of winter temperature, first increasing and then decreasing. During overwintering, the SOD activity remained higher in the branches of 'Tianhong 2' grafted on GM256, Huang 6, and NO.1 and was relatively lower in the branches of 'Tianhong 2' grafted on M9 and ZC9-3.

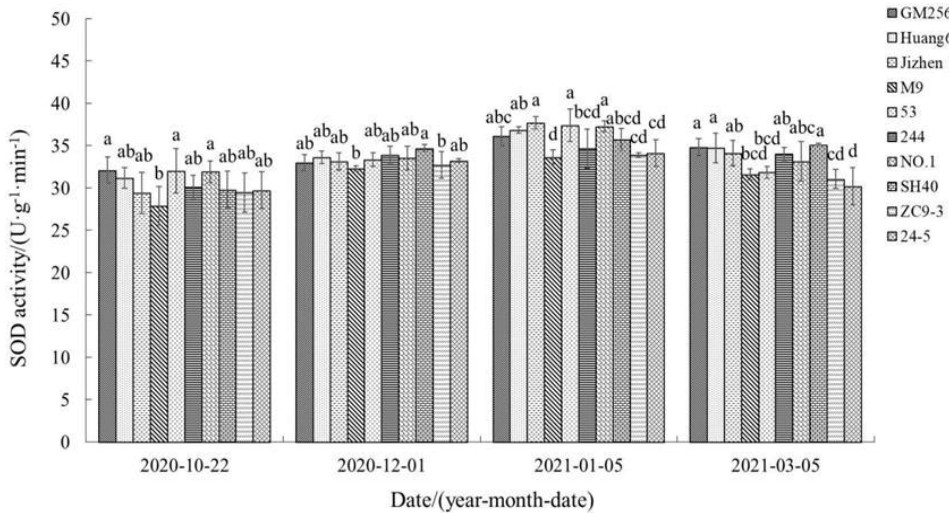

**Figure 3.** SOD activity in 'Tianhong 2' apple branches grafted on different interstocks. Note: In the figure, the lowercase letters indicate that significant differences exist between branches grafted on different interstocks ($p < 0.05$).

### 3.2.3. Change of Peroxide Dismutase (POD) Activity

Like SOD activity, POD activity in 'Tianhong 2' apple branches grafted on different interstocks was opposite to the change trend of winter temperature, first increasing and then decreasing, as shown in Figure 4. Throughout overwintering stress, POD activity remained higher for branches of 'Tianhong 2' grafted on GM256 and Huang 6 and relatively lower in branches of 'Tianhong 2' grafted on M9 and Jizhen 1.

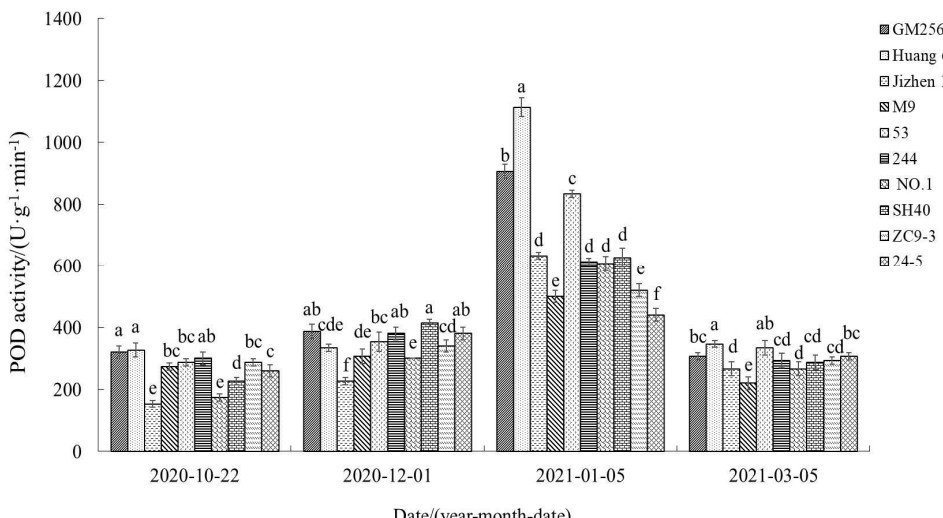

**Figure 4.** POD activity in 'Tianhong 2' apple branches grafted on different interstocks. Note: In the figure, the lowercase letters indicate that significant differences exist between branches grafted on different interstocks ($p < 0.05$).

### 3.2.4. Change of Superoxide Anion ($O_2^-$) Production Rate

As shown in Figure 5, the change of $O_2^-$ production rate of 'Tianhong 2' apple branches grafted on most interstocks was opposite to the change trend of winter temperature, first increasing and then decreasing. Of the tested interstocks, the variation trend of $O_2^-$ production rate in 'Tianhong 2' branches grafted on M9 and 24-5 was slightly different from the others. The $O_2^-$ production rate of branches grafted on M9 continued to increase during the overwintering period but for branches grafted on 24-5 decreased and then increased. These two interstocks exhibited the highest $O_2^-$ production rate in March. The $O_2^-$ production rate remained relatively lower for 'Tianhong 2' branches grafted on GM256 and Huang 6 and relatively higher for branches grafted on M9, Jizhen 1 and ZC9-3.

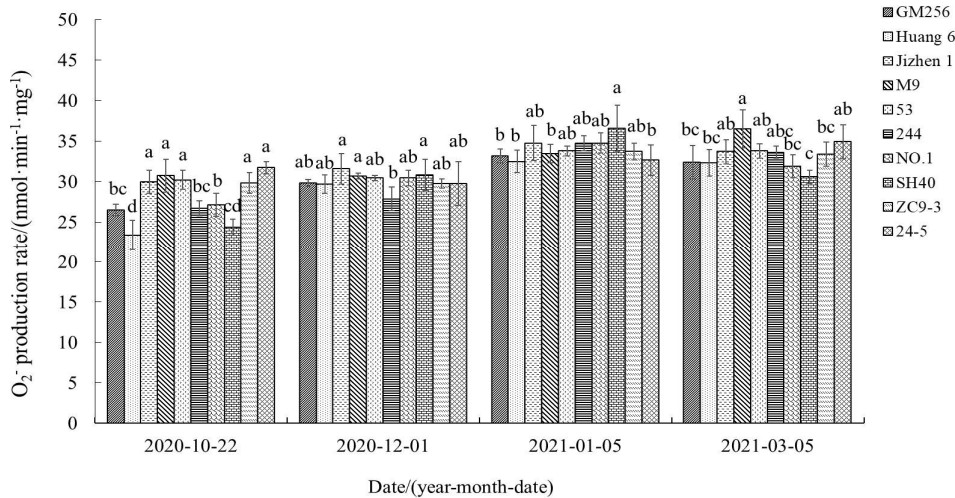

**Figure 5.** Superoxide anion production rate of 'Tianhong 2' apple branches grafted on different interstocks. Note: In the figure, the lowercase letters indicate that significant differences exist between branches grafted on different interstocks ($p < 0.05$).

### 3.2.5. Change of Starch Content

As shown in Figure 6, the content of starch increased and then decreased in 'Tianhong 2' branches grafted on GM256, Huang 6, Jizhen 1, M9, 53, 244, 1, SH40, and ZC9-3 interstocks.

The change trend of starch content in branches grafted on 24-5 was slightly different, with a continuous decreasing trend through the overwintering period. At various times throughout overwintering, there were significant differences in starch content in 'Tianhong 2' apple branches grafted on different rootstocks.

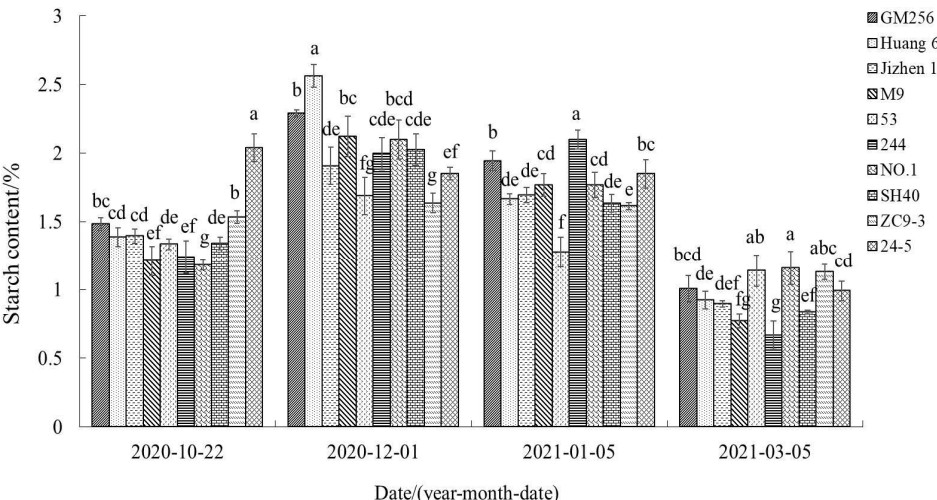

**Figure 6.** Starch content in 'Tianhong 2' apple branches grafted on different interstocks. Note: In the figure, the lowercase letters indicate that significant differences exist between branches grafted on different interstocks ($p < 0.05$).

### 3.2.6. Change of Soluble Sugar Content

As shown in Figure 7, the change of soluble sugar content in 'Tianhong 2' apple branches grafted on different interstocks was opposite to the change trend of winter temperature, first increasing and then decreasing. At all time points during overwintering, the soluble sugar content was at a relatively higher level in 'Tianhong 2' branches grafted on 244 and 24-5 and at a relatively lower level in branches of 'Tianhong 2' grafted on M9 and Jizhen 1.

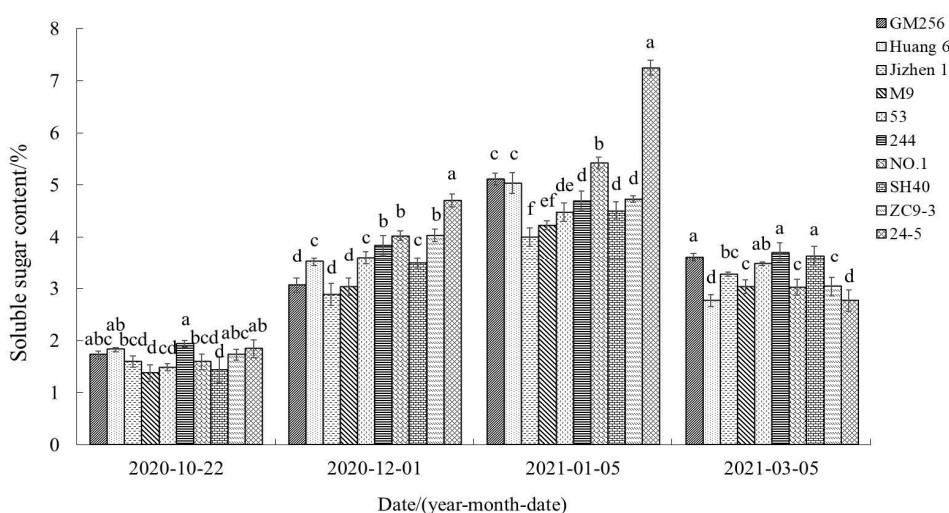

**Figure 7.** Soluble sugar content in 'Tianhong 2' apple branches grafted on different interstocks. Note: In the figure, the lowercase letters indicate that significant differences exist between branches grafted on different interstocks ($p < 0.05$).

### 3.2.7. Change of Soluble Protein Content

As shown in Figure 8, the change of soluble protein content was opposite to the change trend of winter temperature, dramatically increasing and then decreasing slightly. There was significant variation in soluble protein content of the different grafted materials, with relatively lower soluble protein content of 'Tianhong 2' branches grafted on NO.1 throughout the measurement process.

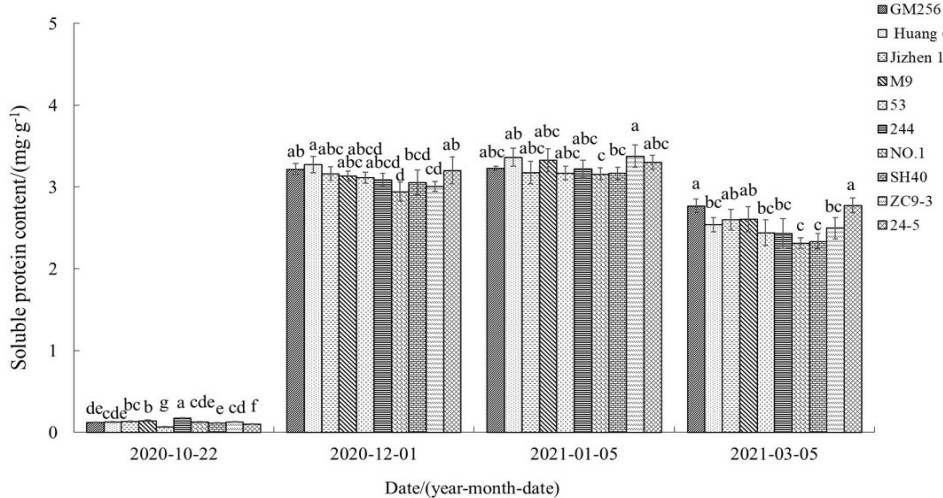

**Figure 8.** Soluble protein content in 'Tianhong 2' apple branches grafted on different interstocks. Note: In the figure, the lowercase letters indicate that significant differences exist between branches grafted on different interstocks ($p < 0.05$).

### 3.2.8. Change of Proline Content

As shown in Figure 9, the proline content changed inversely to winter temperature, first increasing and then decreasing. There were significant differences in 'Tianhong 2' apple branches grafted on the different interstocks for all measurements, with consistently lower proline content of 'Tianhong 2' branches grafted on M9.

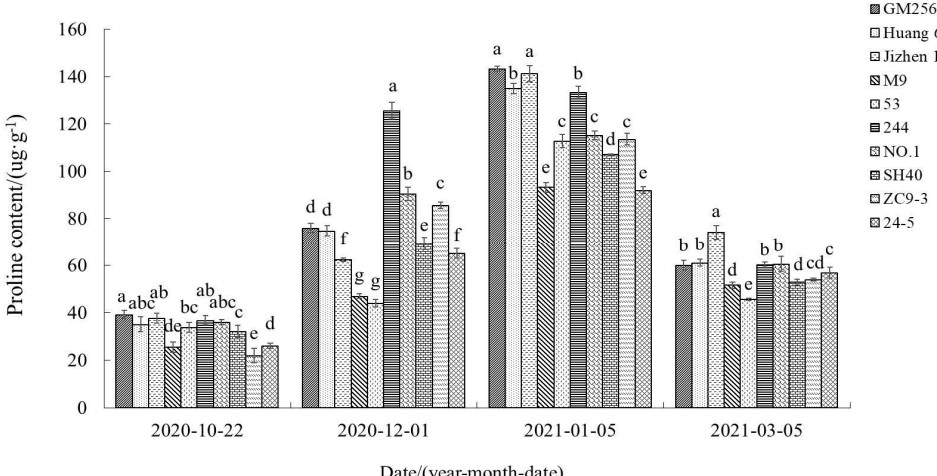

**Figure 9.** Proline content in 'Tianhong 2' apple branches grafted on different interstocks. Note: In the figure, the lowercase letters indicate that significant differences exist between branches grafted on different interstocks ($p < 0.05$).

### 3.3. Comprehensive Evaluation of the Effects of Different Interstocks on the Cold Resistance of 'Tianhong 2' Apple Tree

3.3.1. Correlation Analysis of Physiological Indexes of 'Tianhong 2' Apple Branches Grafted on Different Interstocks

Correlation analysis of physiological indexes of 'Tianhong 2' apple branches grafted on different interstocks was performed, and the results are shown in Table 4. LT50 values were negatively correlated with seven physiological indicators (SOD and POD enzyme activities, the production rate of superoxide anion, the contents of soluble protein, MDA, proline, and soluble sugar). SOD activity was positively correlated with six physiological indicators (POD activity, the production rate of superoxide anion, the contents of soluble protein, MDA, proline, and soluble sugar). POD activity was positively correlated with five physiological indicators (the production rate of superoxide anion, the contents of soluble protein, MDA, proline, and soluble sugar). The production rate of superoxide anion was positively correlated with four physiological indicators (the contents of soluble protein, MDA, proline, and soluble sugar). The soluble protein content was positively correlated with four physiological indicators (the contents of starch, MDA, proline, and soluble sugar). The MDA content was positively correlated with the contents of proline and soluble sugar. There were also positive correlations between starch and proline content, and between proline and soluble sugar content. The correlations between the indexes and the overlap of some information suggest that the evaluation of different interstocks on the cold resistance of 'Tianhong 2' requires multiple indexes. Principal component analysis is a comprehensive evaluation tool that can assess cold resistance by synthesizing the contributions of multiple indexes.

**Table 4.** Correlation Analysis between physiological indexes of 'Tianhong 2' apple branches grafted on different interstocks.

| Correlation Coefficient | LT50 | SOD | POD | $O_2{}^-$ | Protein | MDA | Starch | Proline | Sugar |
|---|---|---|---|---|---|---|---|---|---|
| LT50 | 1 | | | | | | | | |
| SOD | −0.797 ** | 1 | | | | | | | |
| POD | −0.630 ** | 0.713 ** | 1 | | | | | | |
| $O_2{}^-$ | −0.673 ** | 0.455 ** | 0.387 * | 1 | | | | | |
| Protein | −0.931** | 0.722 ** | 0.492 ** | 0.608 ** | 1 | | | | |
| MDA | −0.692 ** | 0.611 ** | 0.507 ** | 0.614 ** | 0.688 ** | 1 | | | |
| Starch | −0.164 | 0.184 | 0.283 | −0.125 | 0.314 * | 0.04 | 1 | | |
| Proline | −0.799 ** | 0.810 ** | 0.807 ** | 0.481 ** | 0.726 ** | 0.605 ** | 0.382 * | 1 | |
| Sugar | −0.847 ** | 0.748 ** | 0.642 ** | 0.541 ** | 0.824 ** | 0.706 ** | 0.308 | 0.790 ** | 1 |

Note: * represents significant difference ($p < 0.05$), ** represents extremely significant different ($p < 0.01$).

3.3.2. Principal Component Factor Analysis of Various Indexes of 'Tianhong 2' Apple Branches Grafted on Different Interstocks

Principal component analysis was performed using SPSS; the principal component eigenvalues (Table 5) and the principal component initial factor load matrix (Table 6) were determined; and the feature vectors were calculated (Table 6). As shown in Table 5, the first two principal components with eigenvalues greater than 1 were selected by SPSS, and the cumulative contribution rate of these two components reached 78.562%, indicating that the first two principal components expressed 78.562% of the cold resistance of 'Tianhong 2' apple branches on 10 interstocks. Therefore, these two principal components can be used as comprehensive indexes to evaluate the influence of 10 interstocks on cold resistance of 'Tianhong 2'. As shown in Table 6, the absolute value of factor load of the first principal component is larger, and the order of absolute value of factor load is as follows: LT50 > soluble sugar > proline > soluble protein > SOD > MDA > POD > $O_2{}^-$. The results showed that principal component 1 mainly reflected LT50, soluble sugar content, proline content, soluble protein content, SOD enzyme activity, MDA content, POD enzyme activity, and superoxide anion production rate. The starch content of principal component 2 accounted

for the largest proportion and was significantly higher than the factor load of principal component 1, indicating that principal component 2 mainly reflected the information of starch content.

**Table 5.** Component eigenvalue.

| Principal Component | Initial Eigenvalue | | |
| --- | --- | --- | --- |
| | Characteristic Root | Contribution Rate (%) | Cumulative Contribution Rate (%) |
| 1 | 5.83 | 64.773 | 64.773 |
| 2 | 1.241 | 13.789 | 78.562 |

**Table 6.** Component Matrix.

| | Load | | Eigenvector | |
| --- | --- | --- | --- | --- |
| Biochemical Indexes | Component 1 | Component 2 | Component 1 | Component 2 |
| LT50 | −0.942 | 0.121 | −0.390 | 0.109 |
| Sugar | 0.911 | 0.056 | 0.377 | 0.050 |
| Proline | 0.902 | 0.226 | 0.374 | 0.203 |
| Protein | 0.896 | −0.025 | 0.371 | −0.022 |
| SOD | 0.869 | 0.053 | 0.360 | 0.048 |
| MDA | 0.787 | −0.291 | 0.326 | −0.261 |
| POD | 0.768 | 0.232 | 0.318 | 0.208 |
| $O_2^-$ | 0.672 | −0.538 | 0.278 | −0.483 |
| Starch | 0.287 | 0.861 | 0.119 | 0.773 |

### 3.3.3. Comprehensive Evaluation and Classification of Cold Resistance of 'Tianhong 2' Apple Branches Grafted on Different Interstocks

The characteristic vector value of each principal component was calculated from the eigenvalue and initial load factor, and then the principal component value was calculated from the normalized original data. The comprehensive score was then obtained according to the contribution rate of each principal component. Finally, the cold resistance was assessed based on the comprehensive scores. Interstocks with high comprehensive score have strong cold resistance, and those with low scores have poor cold resistance. The results are shown in Table 7. From strongest to weakest, the influence of each interstock on the cold resistance of 'Tianhong 2' was in the order of: GM256 > Huang 6 > 244 > NO.1 > 53 > 24-5 > ZC9-3 > Jizhen1 > SH40 > M9.

**Table 7.** Principal component scores, comprehensive scores, ranking, and classification.

| Interstock | Principal Component Value | | Comprehensive Score | Cold Resistance Order (1, Highest to 10, Lowest) | Classification |
| --- | --- | --- | --- | --- | --- |
| | Component 1 | Component 2 | | | |
| GM256 | 1.420503 | 2.516214 | 1.267063 | 1 | Strong tolerance |
| Huang 6 | 0.940823 | 2.989505 | 1.021622 | 2 | Strong tolerance |
| Jizhen1 | −0.28465 | −1.03902 | −0.32764 | 7 | Mean tolerance |
| M9 | −2.81883 | −1.96704 | −2.09708 | 10 | Weak tolerance |
| 53 | 0.784528 | −2.20221 | 0.204499 | 5 | Relatively resistant |
| 244 | 0.554611 | 0.944982 | 0.489542 | 3 | Relatively resistant |
| NO. 1 | 0.660832 | 0.020357 | 0.430848 | 4 | Relatively resistant |
| SH40 | −0.95421 | 0.244786 | −0.58431 | 9 | Mean tolerance |
| ZC9-3 | −0.2112 | −1.60406 | −0.35798 | 8 | Mean tolerance |
| 24-5 | −0.09241 | 0.096477 | −0.04656 | 6 | Relatively resistant |

### 3.4. Cluster Analysis of Cold Resistance of 'Tianhong 2' Apple Branches Grafted on Different Interstocks

Cluster analysis was carried out according to the comprehensive scores of principal component evaluation using Ward's method. The results are shown in Figure 10 and Table 6. The 10 interstocks can be divided into four categories. The first category includes GM256 and Huang 6, with the strongest cold resistance; the second category includes 244, NO.1, 53, and 24-5, with strong cold resistance; the third category includes ZC9-3, Jizhen1, and SH40, with medium cold resistance; and the fourth category includes M9, with weak cold resistance.

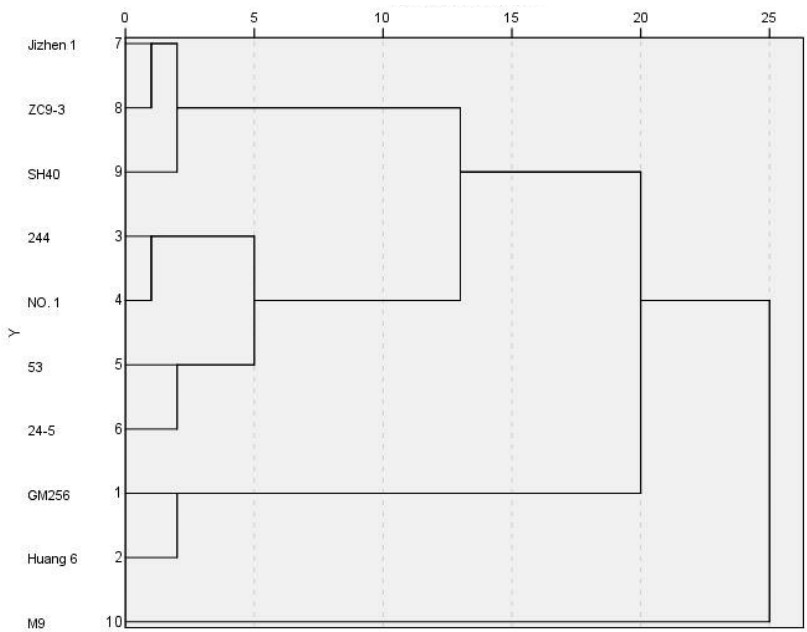

**Figure 10.** Cluster analysis of cold resistance of branches grafted on 10 interstocks.

## 4. Discussion

### 4.1. Relationship between LT50 and Cold Resistance

Under low-temperature stress, the permeability of plant cell membranes increases and intracellular electrolytes leak out, changing the conductivity. Thus, the determination of relative conductivity is an effective method to measure the cold resistance of plants [7], with electrical conductivity negatively correlated with plant cold resistance [24]. Measurements of conductivity can be analyzed by logistic equations to calculate the inflection point temperature and estimate the semi-lethal temperature [25]. LT50 values change with the change of external temperature, and LT50 values should be measured in the coldest month during overwintering to reliably assess apple cold resistance. In this study, the LT50 values of branches of 'Tianhong 2' grafted on GM256, Huang 6, and 244 measured in the coldest month of January were significantly lower than those of other interstocks, indicating strongest cold resistance. The LT50 values of 'Tianhong 2' branches grafted on M9 were the highest, indicating the weakest cold resistance.

### 4.2. Relationship between MDA Content and Cold Resistance

Under low-temperature stress, the degree of cell membrane damage and plant cold resistance can be evaluated by measuring MDA content [26]. The content of MDA in mango increased with the decrease in temperature [27]. In this study, with the decrease in temperature, the MDA content of 'Tianhong 2' apple branches grafted on different interstocks showed a gradual upward trend. MDA is the product of membrane lipid peroxidation, and the increase in MDA content indicates that low-temperature conditions increase reactive oxygen species, which intensifies the peroxidation and plant membrane damage [28]. The

study on cold resistance showed that varieties with stronger cold resistance had higher MDA content, while those with weaker cold resistance had lower MDA content [29]. In this study, throughout the overwintering process, the MDA content was lowest in branches of 'Tianhong 2' grafted on GM256 and Huang 6 and highest in branches of 'Tianhong 2' grafted on ZC9-3.

### 4.3. Relationship between Reactive Oxygen Species, Antioxidant System, and Cold Resistance

Under low-temperature stress, the higher amounts of reactive oxygen species are produced in plants, and the excessive accumulation of reactive oxygen species will lead to membrane phospholipid peroxidation and damage to the plant [30]. SOD and POD are important components of the enzymatic defense system and important protective enzymes in plants. These enzymes can remove reactive oxygen species produced during cold injury, reduce stress damage, and protect normal plant growth [31,32]. A previous study showed that SOD and POD enzyme activities were highest and superoxide anion content was lowest in varieties with strong cold resistance after low-temperature stress treatment; the content of superoxide anion in varieties with poor cold resistance was always at a high level [33]. In this study, SOD and POD enzyme activities were higher and superoxide anion production rate was lower in rootstock-interstock-scion combinations with stronger cold resistance. SOD and POD activities were lower and superoxide anion production rate was higher in rootstock-interstock-scion combinations with poorer cold resistance. This may indicate that plants with higher protective enzyme activity can remove more reactive oxygen species in vivo, thus reduce damage to the membrane system caused by low temperature and enhancing cold tolerance of plants.

### 4.4. Relationship between the Content of Osmotic Adjustment Substancse and Cold Resistance

Soluble sugar, soluble protein, and proline are common osmoregulatory substances in plants [14]. The accumulation of soluble sugar can protect plants from low-temperature freezing injury and reduce the damage to plants. When subjected to low-temperature stress, plants can increase the soluble sugar content to resist the effects of the cold [34]. In this study, the soluble sugar content in all apple branches increased with the decrease in temperature and decreased with the increase in temperature. Albina found that low-temperature stress can cause a change in the soluble protein content in cells, and the change of soluble protein content is closely related to cold resistance [35]. In this study, with the decrease in temperature, the soluble protein content in all apple branches increased, and there were significant variations between the contents of soluble protein in 'Tianhong 2' apple branches grafted on different interstocks, affecting their cold resistance. Proline enhances plant cold resistance by increasing plant water retention capacity [14]. In this study, with the decrease in temperature, the proline content increased in 'Tianhong 2' apple branches grafted on all different interstocks. The proline content was higher in branches of rootstock-interstock-scion combinations with stronger cold resistance. This result is consistent with the findings of Zhang et al. [19].

### 4.5. Relationship between Starch Content and Cold Resistance

In this study, the starch content first increased and then decreased with the decrease in temperature. In the early stage, starch accumulates in plants, so the starch content showed an increasing trend. With low-temperature stress, plants transformed starch into soluble sugar to enhance cold resistance, so the starch content decreased. This is consistent with the findings of Bertrand et al. [36], that the starch content of alfalfa decreases with the progress of low-temperature acclimation. However, other studies did not observe a decrease in starch content in response to low-temperature changes [37], suggesting that this process may depend on the characteristics of different plant species.

## 5. Conclusions

The tested interstocks exhibited different effects on the cold resistance of 'Tianhong 2' apple. The apple interstocks exhibited cold resistance in the order of: GM256 > Huang 6 > 244 > NO.1 > 53 > 24-5 > ZC9-3 > Jizhen1 > SH40 > M9. The cold resistance of these 'Tianhong 2' interstocks can be divided into four categories. GM256 and Huang 6 exhibit the strongest cold resistance; 244, NO.1, 53, and 24-5 exhibit strong cold resistance; ZC9-3, Jizhen1, and SH40, exhibit medium cold resistance; and M9 showed weak cold resistance.

**Author Contributions:** J.J. wrote the paper, M.L. carried out the experiment, B.Y. conducted the data analysis, B.L. and Z.L. managed the material, J.X. provided the material, X.Z. and S.Z. designed the experiment. All authors have read and agreed to the published version of the manuscript.

**Funding:** This research was funded by the National Natural Science Foundation of China (32002008), the Natural Science Foundation of Hebei Province (C2020204015), the Key Research and Development Program of Hebei Province (19226317D), Basic scientific research funds for colleges and universities in Hebei Province (KY2021058), and the Key Research and Development Plan Project of Hebei Province (21326308D-02-03).

**Data Availability Statement:** Data sharing not applicable: No new data were created or analyzed in this study. Data sharing is not applicable to this article.

**Acknowledgments:** We thank Zhixue Hu for the grafting of the apple trees.

**Conflicts of Interest:** The authors have no conflicting interests, and all authors have approved the manuscript and agree with its submission to Horticulturae.

## Abbreviations

| | |
|---|---|
| MDA | Malondialdehyde |
| REL | Relative electrolyte leakage |
| LT50 | Semi-lethal temperature |
| SOD | Superoxide dismutase |
| POD | Peroxidase |
| $\cdot O_2^-$ | Superoxide anion free radical |
| TBA | Thiobarbituric acid |
| NBT | Nitroblue tetrazolium |

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
