# Peer review of "Effects of 10 Dwarfing Interstocks on Cold Resistance of ‘Tianhong 2’ Apple"

_horticulturae, doi:10.3390/horticulturae9070827_

Round 1

Reviewer 1 Report

The manuscript No. horticulturae-2453298 contains very important piece of research work presented satisfactorily. However, it needs further improvement in the syntax to make it more interesting to the readers. The references cited in the manuscript need improvement e.g. references cited in the introduction don’t auger well for the context.

“Apple is one of the most widely cultivated fruit trees in temperate zones and is consumed worldwide providing an important food source [1]. The Chinese apple industry 26 contributes significantly to China's rural economy [2].”

 Reference cited:

1. Chen, X.; Li, S.; Zhang, D.; Han, M.; Jin, X.; Zhao, C.; Wang, S.; Xing, L.; Ma, J.; Ji, J.; An, N. Sequencing of a Wild Apple (Malus baccata) Genome Unravels the Differences Between Cultivated and Wild Apple Species Regarding Disease Resistance and Cold Tolerance. G3-Genes Genom. Genet. 2019, 9(7), 2051-2060. 430

2. Wei, X.; Liu, F.; Chen, C.; Ma, F.; Li, M. The Malus domestica sugar transporter gene family: identifications based on genome and expression profiling related to the accumulation of fruit sugars. Front. Plant Sci. 2014, 5.

Similarly at other places in the manuscript the most relevant reference should be cited.

The manuscript should be accepted after minor revision.

English language is fine. However, it can be further improved by copy editing by the editor.

Author Response

Thanks for your valuable comment. This view is very helpful to improve this paper. We are sorry for these mistakes. We have updated the references, and the  updated references were marked with red in the manuscript.

Reviewer 2 Report

The Abstract needs to be completely revised and written more precisely. The methods used are not clear enough. The type of plant material used from the apple tree must be redescribed. The low-temperature treatment used needs to be described in more detail. In addition, the use of master's theses as references describing the method is not appropriate because they are not sufficiently visible on the Internet. Moreover, I suspect that the author of the master's thesis developed the method mentioned. A similar objection applies to the description of the methods for determining other physiological indices. In the chapters "Results" and "Discussion" the interpretation of statistical significance was not sufficiently made. In some references, the year of publication was written twice.

English is good minor revision is need 

Author Response

Comment 1: The abstract needs to be completely revised and written more precisely.

Response 1: Thanks for your valuable comment. This view is very helpful to improve  this paper. We have revised the abstract and marked the modification with red in the abstract.

Comment 2: The methods used are not clear enough. The type of plant material used from the apple tree must be redescribed. The low-temperature treatment used needs to be described in more detail. In addition, the use of master's theses as references describing the method is not appropriate because they are not sufficiently visible on the Internet. Moreover, I suspect that the author of the master's thesis developed the method mentioned. A similar objection applies to the description of the methods for determining other physiological indices.

Response 2: Thanks for your valuable comment. We have redescribed the type of plant material used from the apple tree, and described the low-temperature treatment used in more detail. We also replaced the master's theses with other references describing the method, and also replaced the references with others in the description of the methods for determining other physiological indices in section 2.2.3. We have marked the modification with red in the manuscript.

Comment 3: In the chapters "Results" and "Discussion" the interpretation of statistical significance was not sufficiently made.

Response 3: Thanks for your valuable comment. The interpretation of statistical significance in the chapters "Results" and "Discussion" was have been modified and marked with red.

Comment 4: In some references, the year of publication was written twice.

Response 4: Thanks for your helpful suggestion. We are sorry for this mistake. We have corrected this mistake and marked with red. (Reference 33)

Reviewer 3 Report

The manuscript entitled "Effects of 10 Dwarfing Interstocks on Cold Resistance of 'Tianhong 2' Apple" presents a study on the effect of rootstock on increasing the cold resistance of an apple tree. The authors associate frost resistance with a number of physiological parameters. The study is of interest to ensure efficient apple production and for breeders.

However, there are a number of remarks to the work.

- Why weren't measurements taken in February?

- For a more accurate identification of the effect of intermediate rootstocks, it would be necessary to include another option in the experiment - the Tianhong 2 variety without an intermediate insert.

Author Response

Comment 1: Why weren't measurements taken in February?

Response 1: Because it was during the winter holiday in February, and it was unable to sample because the school was shut. But that didn't affect the results, because the temperature was lowest in January. Sampling and testing in January were most important to the results. After January, the temperature began to rise, and the tests in February and March both reflected the situation after the temperature rose, so the lack of tests in February did not affect the test results.

Comment 2: For a more accurate identification of the effect of intermediate rootstocks, it would be necessary to include another option in the experiment - the Tianhong 2 variety without an intermediate insert.

Response 2: Thanks for your valuable comment. Your advice is quite correct. We will take this into account in future experiments. In this study, three interstocks with different degrees of cold resistance which were commonly used in production (GM256- with strongest cold resistance, SH40- with stronger cold resistance, M9- with poor cold resistance) were used as control.

Round 2

Reviewer 2 Report

The research paper entitled: 'Effects of 10 Dwarfing Interstocks on Cold Resistance of 'Tianhong 2' Apple' is significantly improved. The methods used are described much better. I propose to publish this paper in a journal `Horticulture` without further revision.